# A DFT Characterization of Structural, Mechanical, and Thermodynamic Properties of Ag$_9$In$_4$ Binary Intermetallic Compound

**Hsien-Chie Cheng [1] and Ching-Feng Yu [2],***

1   Department of Aerospace and Systems Engineering, Feng Chia University, Taichung 40724, Taiwan
2   Electronic and Optoelectronic System Research Laboratories, Industrial Technology Research Institute (ITRI), Hsinchu 30010, Taiwan
*   Correspondence: d9733548@oz.nthu.edu.tw

**Abstract:** The intermetallic compounds (IMCs) at the interface between the solder joint and metal bond pad/under bump metallization (UBM) exert a significant impact on the thermal–mechanical behavior of microelectronic packages because of their unique physical properties. In this study, a theoretical investigation of the physical properties, namely structural, mechanical, and thermodynamic properties, of the Ag$_9$In$_4$ IMC was conducted using ab initio density functional theory (DFT) calculations. The calculated equilibrium lattice constants were in good agreement with the literature experimental data. Furthermore, with the calculated elastic constants, we can derive the ductility and brittleness nature, elastic anisotropy, and direction-dependent elastic properties of Ag$_9$In$_4$ through several elastic indices, three-dimensional surface representation, and two-dimensional projections of elastic properties. The calculations inferred that the cubic Ag$_9$In$_4$ IMC confers structural and mechanical stability, ductility, relative low stiffness and hardness, and elastic anisotropy. Finally, the thermodynamic properties, i.e., Debye temperature, heat capacity, and minimum thermal conductivity, were also investigated. Evidently, the low-temperature heat capacity conforms to the Debye heat capacity theory and the high-temperature one complies with the classical Dulong–Petit law.

**Keywords:** intermetallic compound; ab initio calculations; density functional theory; structure-property; mechanical properties; thermodynamic properties

## 1. Introduction

In recent years, a great demand for electromechanical devices capable of operating at high temperatures has dramatically arisen. For instance, hybrid electric vehicles (HEVs) integrating a gasoline engine with an electric motor are equipped with built-in inverters to control the flow of electric current [1]. These inverters comprise power modules with an array of power integrated circuits (ICs) for low power loss, high current/voltage, high switching frequency, and high-temperature operation [2]. Solders with high content of lead (Pb) are generally adopted as die attachment materials for power semiconductor packaging. However, Pb is an extremely deadly chemical, and, if not disposed of appropriately, would cause severe damage to human health and the environment. For example, even low-level exposure to Pb may still induce considerable deleterious chronic diseases in humans. Recently, Pb has been banned from use in electronics devices, and many innovative Pb-free solders have been reported as promising alternatives to Pb-based solders [3–5], where a number of metal elements, including indium (In), bismuth (Bi), antimony (Sb), and zinc (Zn), are used to replace Pb in the solder paste. Among these alternatives, In-based and pure In solders are very attractive because of their excellent wettability, low melting temperature, great thermal fatigue resistance, and good impact resistance [6–8]. These make them a very encouraging prospect for low-temperature bonding and high interconnect reliability in microelectronic packaging [9].

Many fundamental technical issues need to be addressed prior to the application of In-based solders. For example, Ag alloy has been widely used in microelectronic packaging as under bump metallurgy (UBM) [10,11]. During the solder joint fabrication process, the Ag-based UBM tends to readily dissolve in the In-based solder, eventually bringing about the formation of a very thin $Ag_9In_4$ intermetallic compound (IMC) layer [12]. Song et al. [13] investigated the phase evolution of Ag-In phase IMCs that are obtained from isothermal reactions between In and Ag substrate metal. They indicated that the first emerging IMC phase was $AgIn_2$, but it eventually and completely turned into $Ag_9In_4$ after heating for 30 min at 180 °C. As a result, $Ag_9In_4$ is the primary IMC phase among the Ag-In phase IMCs [13–15]. IMCs can give rise to a huge impact on the structural strength and material properties of solder interconnects, which powerfully correlate with the thermal–mechanical behavior of microelectronic packages. For instance, Qin et al. [16] reported that the growth in the IMCs' thickness above a threshold point would elevate the risk of interconnect damage under loads. Cheng et al. [17] also found that the drop impact interconnect fatigue life of three-dimensional (3D) chip-on-chip packaging would reduce with an increasing IMC thickness. In brief, the thickness of IMCs plays a significant role in the mechanical strength and fatigue life of the solder interconnects. Over the years, several studies have focused on the formation and evolution of the $Ag_9In_4$ IMC [14,18,19]. Before effective implementation of In-based solders in microelectronics packaging, a thorough understanding of its physical characteristics is vital in successfully grasping the thermal fatigue life of the solder interconnects. Investigations of the material properties of the $Ag_9In_4$ crystal were extremely limited until Inukai et al. [20], where the electronic structure of $Ag_9In_4$ was explored through first-principles simulations. Song et al. [13] explored the Young's modulus and hardness of the $Ag_9In_4$ IMC by an experimental approach using nanoindentation. In contrast, its mechanical and thermodynamic properties are hardly addressed in the literature, and worth further investigation. Thus, this study aimed to conduct a theoretical estimation of the physical properties of the $Ag_9In_4$ IMC, comprising structural, mechanical, and thermodynamic properties, using ab initio density functional theory (DFT) calculations [21–23]. The focus was placed on the theoretical estimation of its physical characteristics, such as mechanical stability, ductility and brittleness nature, elastic isotropy and anisotropy, and directional relationship of elastic properties.

## 2. Computational Method and Details

The $Ag_9In_4$ crystal system is cubic with P-43M space group with a = b = c = 9.922 Å, wherein the atomic locations of Ag and In atoms in an elementary cell are depicted in Table 1 [24]. Figure 1 further exhibits the crystal structure of $Ag_9In_4$. In the study, the ab initio study based on DFT [22–24] by Cambridge Serial Total Energy Package (CASTEP) (version 2020, BIOVIA, San Diego, CA, USA) [25] was carried out to evaluate the physical properties of the $Ag_9In_4$ IMC. For years, this package has been extensively utilized to analyze the physical properties and behaviors of several crucial materials [26–34]. For example, Pan. [28] explored the structural stability and optical properties of NiPt nano-material using DFT calculations. Yu et al. [29] examined the elasticity, sound velocity, and minimum thermal conductivity of low-boride VxBy compounds using first-principles calculations. Lin et al. [30] applied first-principles calculations to investigate the thermal and electrical transport properties of bcc and fcc dilute Fe–X (X = Al, Co, Cr, Mn, Mo, Nb, Ni, Ti, V, and W) binary alloys. Moreover, Li et al. [33] employed DFT methods to study the elastic and thermal properties of $M_2InX$ (M = Ti, Zr and X = C, N) phases. An ultrasoft pseudopotential [35] was applied to describe the interactions of ion–electron, and the exchange–correlation energy was approximated by the Perdew–Burke–Ernzerhof (PBE) generalized gradient approximation (GGA) [22]. Moreover, the ground state configuration of the crystal was derived through energy minimization using the Broyden–Fletcher–Goldfarb–Shanno (BFGS) quasi-Newton algorithm [36]. Moreover, the selected plane wave basis set cut-off was 400 eV. The Brillouin-zone sampling [37] adopted an 8 × 8 × 8 k-point mesh generated by the Monkhorst–Pack algorithm [38]. In the geometry optimization, the

following convergence thresholds were employed: the maximum energy change, force, stress and ionic displacement were set to $5 \times 10^{-6}$ eV/atom, 0.01 eV/Å, 0.02 GPa, and $5 \times 10^{-4}$ Å.

**Table 1.** Atomic coordinates for $Ag_9In_4$ before relaxation.

| Atom | $x$ | $y$ | $z$ |
|------|-----|-----|-----|
| Ag | 0.605 | 0.605 | 0.605 |
| Ag | −0.162 | −0.162 | −0.162 |
| Ag | 0.318 | 0.318 | 0.318 |
| Ag | 0 | 0 | 0.355 |
| Ag | 0.5 | 0.5 | 0.854 |
| Ag | 0.320 | 0.320 | 0.034 |
| In | 0.122 | 0.122 | 0.122 |
| In | 0.808 | 0.808 | 0.530 |

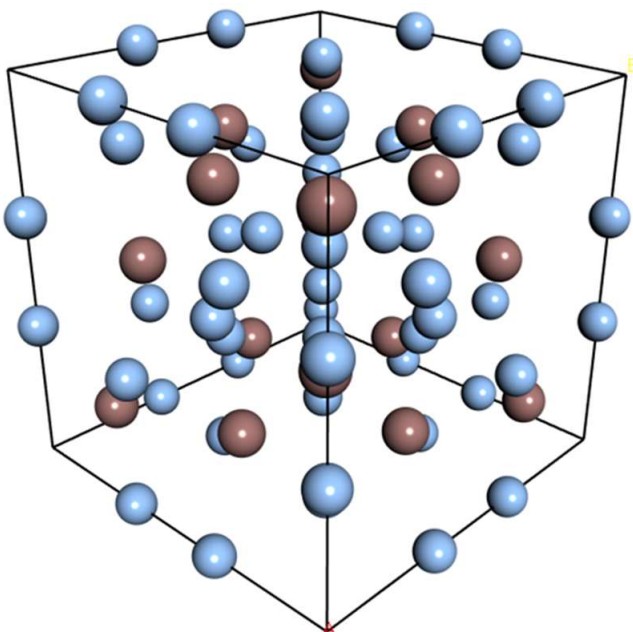

**Figure 1.** Crystal structure of $Ag_9In_4$ (The spheres in blue are Ag atoms and the spheres in brown are In atoms).

Generally, the elastic coefficients are closely related to the fundamental microscopic and macroscopic physical properties of materials, by which the information of crystal stability and stiffness can be disclosed. The elastic coefficients of materials can be derived by the generalized Hooke's law:

$$\sigma_{ij} = C_{ijkl}\varepsilon_{kl} \tag{1}$$

where $\sigma_{ij}$ and $\varepsilon_{kl}$ denote the Cauchy stress tensor ($\sigma_{ij} = \sigma_{ji}$) and infinitesimal strain tensor, respectively, and $C_{ijkl}$ is the elastic stiffness tensor. It is often useful to express Hooke's law in matrix notation, also called Voigt notation. Then, by taking advantage of the symmetry of the stress and strain tensors and expressing them as six-dimensional vectors in an orthonormal coordinate system ($e_1$, $e_2$, $e_3$) as

$$[\sigma] = \begin{bmatrix} \sigma_{11} \\ \sigma_{22} \\ \sigma_{33} \\ \sigma_{23} \\ \sigma_{13} \\ \sigma_{12} \end{bmatrix} = \begin{bmatrix} \sigma_1 \\ \sigma_2 \\ \sigma_3 \\ \sigma_4 \\ \sigma_5 \\ \sigma_6 \end{bmatrix} ; [\varepsilon] = \begin{bmatrix} \varepsilon_{11} \\ \varepsilon_{22} \\ \varepsilon_{33} \\ 2\varepsilon_{23} \\ 2\varepsilon_{13} \\ 2\varepsilon_{12} \end{bmatrix} = \begin{bmatrix} \varepsilon_1 \\ \varepsilon_2 \\ \varepsilon_3 \\ \varepsilon_4 \\ \varepsilon_5 \\ \varepsilon_6 \end{bmatrix} \tag{2}$$

Then, the stiffness tensor (C) can be expressed as

$$[C] = \begin{bmatrix} C_{1111} & C_{1122} & C_{1133} & C_{1123} & C_{1131} & C_{1112} \\ C_{2211} & C_{2222} & C_{2233} & C_{2223} & C_{2231} & C_{2212} \\ C_{3311} & C_{3322} & C_{3333} & C_{3323} & C_{3331} & C_{3312} \\ C_{2311} & C_{2322} & C_{2333} & C_{2323} & C_{2331} & C_{2312} \\ C_{3111} & C_{3122} & C_{3133} & C_{3123} & C_{3131} & C_{3112} \\ C_{1211} & C_{1222} & C_{1233} & C_{1223} & C_{1231} & C_{1212} \end{bmatrix} = \begin{bmatrix} C_{11} & C_{12} & C_{13} & C_{14} & C_{15} & C_{16} \\ C_{12} & C_{22} & C_{23} & C_{24} & C_{25} & C_{26} \\ C_{13} & C_{23} & C_{33} & C_{34} & C_{35} & C_{36} \\ C_{14} & C_{24} & C_{34} & C_{44} & C_{45} & C_{46} \\ C_{15} & C_{25} & C_{35} & C_{45} & C_{55} & C_{56} \\ C_{16} & C_{26} & C_{36} & C_{46} & C_{56} & C_{66} \end{bmatrix} \tag{3}$$

and Hooke's law is written as

$$\{\sigma_i\} = [C_{ij}]\{\varepsilon_i\} \tag{4}$$

For a cubic system, the $C_{ij}$ can be described as [39]

$$[C_{ij}] = \begin{bmatrix} C_{11} & C_{12} & C_{12} & 0 & 0 & 0 \\ & C_{11} & C_{12} & 0 & 0 & 0 \\ & & C_{11} & 0 & 0 & 0 \\ & & & C_{44} & 0 & 0 \\ & \text{symm.} & & & C_{44} & 0 \\ & & & & & C_{44} \end{bmatrix} \tag{5}$$

The matrix has only three independent elastic coefficients (i.e., $C_{11}$, $C_{12}$, and $C_{44}$). The inverse of the elastic stiffness tensor $C_{ij}$ yields the compliance tensor.

To evaluate the elastic anisotropic features of the cubic Ag$_9$In$_4$ crystal, a 3D surface construction of the elastic anisotropy was carried out. The directional relationship of the Young's modulus and shear modulus of the Ag$_9$In$_4$ crystal were explored using the following equations [40],

$$E(\theta, \varphi) = \frac{1}{S'_{11}(\theta, \varphi)} = \frac{1}{a_i a_j a_k a_l S_{ijkl}} \tag{6}$$

$$G(\theta, \varphi, \chi) = \frac{1}{4S'_{66}(\theta, \varphi, \chi)} = \frac{1}{4a_i b_j a_k b_l S_{ijkl}} \tag{7}$$

where $S_{ijkl}$ stands for the compliance coefficients and $a$ and $b$ are the unit vectors. The unit vector a needs two angles $\theta$ and $\varphi$ to describe it. In addition, shear modulus requires another unit vector $b$, which is characterized by the angle $\chi$. Moreover, unit vector $b$ is perpendicular to unit vector $a$. The relationships of unit vector $a$, unit vector $b$ and angles $\theta$, $\varphi$ and $\chi$ are shown in Figure 2. The $\theta$ is in the range of $0 \sim \pi$; $\varphi$ and $\chi$ are in range of $0 \sim 2\pi$. The coordinates of two vectors $a$ and $b$ are shown below.

$$a = \begin{pmatrix} \sin\theta\cos\varphi \\ \sin\theta\sin\varphi \\ \cos\theta \end{pmatrix}, \text{and } b = \begin{pmatrix} \cos\theta\cos\varphi\cos\chi - \sin\theta\sin\chi \\ \cos\theta\sin\varphi\cos\chi + \cos\theta\sin\chi \\ -\sin\theta\cos\chi \end{pmatrix} \tag{8}$$

The calculation results of phonon spectra were used to compute heat capacity ($C_v$) versus temperature [41]. The temperature dependence of the energy can be calculated as follows:

$$E(T) = E_{tot} + E_{zp} + \int \frac{\hbar\omega}{\exp(\frac{\hbar\omega}{kT}) - 1} F(\omega) d\omega \tag{9}$$

where $E_{tot}$ stands for the total static at 0 K, $E_{zp}$ is the zero-point vibrational energy, $\omega$ is the frequency, $k$ is the Boltzmann constant, $T$ is the Kelvin temperature, $\hbar$ is Planck's constant and $F(\omega)$ is the phonon density of states. $E_{zp}$ can be expressed as follows:

$$E_{zp} = \frac{1}{2} \int F(\omega)\hbar\omega d\omega \tag{10}$$

The lattice contribution to the heat capacity, $C_V$, is calculated as follows:

$$C_v = k \int \frac{\left(\frac{\hbar\omega}{kT}\right)^2 \exp\left(\frac{\hbar\omega}{kT}\right)}{\left[\text{xp}\left(\frac{\hbar\omega}{kT}\right) - 1\right]^2} F(\omega)d\omega \tag{11}$$

A well-known representation of the experimental data on heat capacity is based on the comparison of the actual heat capacity to that predicted by the Debye model. This leads to the concept of the temperature-dependent Debye temperature, $\theta_D$. Heat capacity in the Debye model is given by [42]:

$$C_v(t) = 9Nk \left(\frac{t}{\theta_D}\right)^3 \int_0^{\theta_D/t} \frac{x^4 e^x}{(e^x - 1)^2} dx \tag{12}$$

where $N$ is the number of atoms per cell. There, the value of the Debye temperature, $\theta_D$, at a given temperature, $t$, is obtained by calculating the heat capacity, Equation (8), then inverting Equation (9) to obtain $\theta_D$.

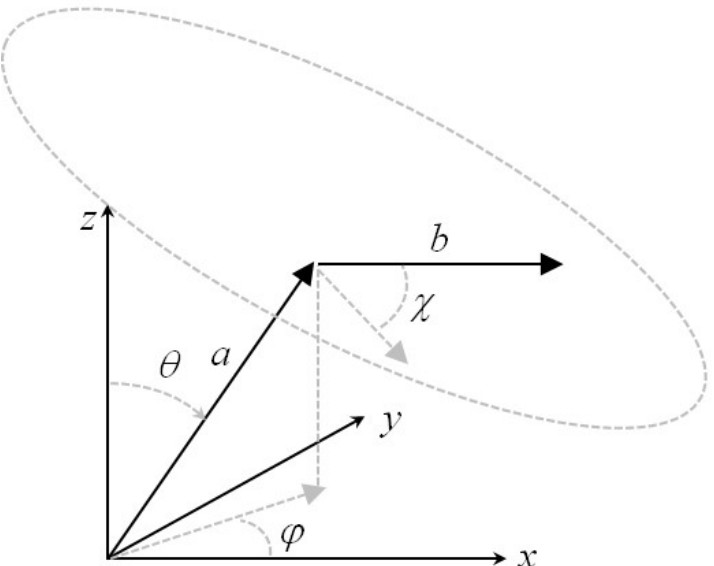

**Figure 2.** Definitions of angles used to describe directions in calculations.

## 3. Results and Discussion

### 3.1. Structural Properties

First of all, the equilibrium lattice parameters and relaxed atom structure of the $Ag_9In_4$ unit cell in the ground state were computed through structural optimization. Table 2 shows the relaxed atomic coordinates of the $Ag_9In_4$ crystal. The calculated lattice constants and volume of the unit cell together with the literature data [24] are exhibited in Table 3. It is clear to see that the modeled results were in a good consistency with the literature data [24] in both length and volume, where the differences in lattice constant and volume are merely about 0.07% and 0.24%, respectively. In this study, the elastic constants of $Ag_9In_4$ were obtained by linear fitting using four strains of $\pm 0.001$ and $\pm 0.003$.

**Table 2.** Atomic coordinates for $Ag_9In_4$ after relaxation.

| Atom | $x$ | $y$ | $z$ |
|---|---|---|---|
| Ag | 0.6049 | 0.6049 | 0.6049 |
| Ag | −0.1699 | −0.1699 | −0.1699 |
| Ag | 0.3259 | 0.3259 | 0.3259 |
| Ag | 0 | 0 | 0.3563 |
| Ag | 0.5 | 0.5 | 0.8556 |
| Ag | 0.3179 | 0.3179 | 0.02834 |
| In | 0.1236 | 0.1236 | 0.1236 |
| In | 0.8103 | 0.8103 | 0.5331 |

**Table 3.** Calculated lattice constants, equilibrium volume $V$, elastic coefficients, Cauchy pressure (GPa), bulk modulus $K$ (GPa), shear modulus $G$ (GPa), Young's modulus $E$ (GPa), hardness $H_v$ (GPa), Poisson's ratio $v$, and ratio of bulk modulus to shear modulus $K/G$ of the $Ag_9In_4$ crystal, together with comparison with the other seven IMC systems in terms of $E$ and $H_v$.

| Mechanical Properties | Lattice Constants (Å) | $V(Å^3)$ | $C_{11}$ | $C_{44}$ | $C_{12}$ | $P^{Cauchy}$ | $K$ | $G$ | $E$ | $H_v$ | $v$ | $K/G$ |
|---|---|---|---|---|---|---|---|---|---|---|---|---|
| Present study | $a = b = c = 10.029$ | 1008.73 | 128.6 | 28.2 | 54.9 | 26.7 | 79.47 | 31.37 | 83.17 | 3.67 | 0.3 | 2.53 |
| Experiment [24] | $a = b = c = 10.037$ | 1011.14 | - | - | - | - | - | - | - | - | - | - |
| Experiment [13] | - | - | - | - | - | - | - | - | 89.4 ± 2.2 | 3.2 ± 0.22 | - | - |
| $Cu_6Sn_5$ [13] | - | - | - | - | - | - | - | - | 110 ± 3.1 | 7.3 ± 0.08 | - | - |
| $Cu_3Sn$ [13] | - | - | - | - | - | - | - | - | 140 ± 2.1 | 7.2 ± 0.21 | - | - |
| $Cu_5Zn_8$ [13] | - | - | - | - | - | - | - | - | 170 ± 2.2 | 6.9 ± 0.09 | - | - |
| $Ni_3Sn_4$ [13] | - | - | - | - | - | - | - | - | 127 ± 2.9 | 4.5 ± 0.13 | - | - |
| $AuZn_3$ [13] | - | - | - | - | - | - | - | - | 111 ± 2.3 | 2.2 ± 0.07 | - | - |
| $Cu_{11}In_9$ [13] | - | - | - | - | - | - | - | - | 109.5 ± 0.84 | 6.57 ± 0.15 | - | - |

### 3.2. Elastic and Mechanical Properties

It is important to note that the calculated elastic coefficients for the crystal must be positive, and obey the mechanical stability criteria [39]:

$$C_{11} - C_{12} > 0, \ C_{11} > 0, \ C_{44} > 0, \ C_{11} + 2C_{12} > 0 \tag{13}$$

The calculated elastic coefficients of the cubic $Ag_9In_4$ crystal structure are tabulated in Table 3, where $C_{11}$, $C_{44}$ and $C_{12}$ are about 128.6 GPa, 28.2 GPa and 54.9 GPa. These elastic coefficients are further used in Equation (13). It turns out that they well conform to these mechanical stability criteria, suggesting that the $Ag_9In_4$ crystal is mechanically stable.

The ductile/brittle nature of a cubic material can be depicted by the Pettifor criterion of Cauchy pressure [43]:

$$P^{Cauchy} = C_{12} - C_{44} \tag{14}$$

Normally, materials having a positive Cauchy pressure reveal metallic-like bonds and ductility. In contrast, a negative Cauchy pressure suggests a brittle material. According to Ref. [44], a negative Cauchy pressure is primarily attributable to the long-range electrostatic contribution. The calculated Cauchy pressure for the $Ag_9In_4$ crystal is given in Table 1. There is an evident positive Cauchy pressure (i.e., 26.7 GPa) for the crystal, thereby exhibiting ductility.

Bulk modulus ($K$) is a measure of the resistance of a material against volume change upon an applied hydrostatic pressure; shear modulus is an estimate of the resistance of a material to shear deformation; Young's modulus is a measure of the resistance of a

material against elastic deformation under load; and Poisson's ratio is a measure of the change in shape of a material to an applied load or the ratio of transverse-to-axial strain of a material under axial load, ranging from $-0.5$ to $0.5$. An increased Poisson's ratio would raise plasticity. These polycrystalline mechanical properties are vital for engineering applications, which can be described as a function of these independent elastic coefficients according to the Voigt–Reuss method [45]:

$$G_V = \frac{1}{5}[C_{11} - C_{12} + 3C_{44}] \tag{15}$$

$$G_R = \frac{15}{4S_{11} - 4S_{12} + 3S_{44}} \tag{16}$$

$$K_V = \frac{1}{3}(C_{11} + 2C_{12}) \tag{17}$$

$$K_R = \frac{1}{3S_{11} + 6S_{12}} \tag{18}$$

where $G_V$ and $G_R$ denote the upper (Voigt) and lower (Reuss) bounds of the shear modulus ($G$) and $K_V$ and $K_R$ denote those of bulk modulus of polycrystalline aggregate, respectively. Furthermore, the effective bulk and shear moduli can be further computed through the Voigt–Reuss–Hill estimate [46], which is roughly equivalent to the geometric mean of bulk and shear moduli:

$$K = \sqrt{K_V \cdot K_R} \tag{19}$$

$$G = \sqrt{G_V \cdot G_R} \tag{20}$$

In addition, with these two moduli, we can obtain the effective Young's modulus and Poisson's ratio of the $Ag_9In_4$ polycrystalline aggregate,

$$E = \frac{9KG}{3K + G} \tag{21}$$

$$v = \frac{3K - 2G}{2(3K + G)} \tag{22}$$

The calculated bulk modulus, shear modulus, Young's modulus, and Poisson's ratio are illustrated in Table 3. The Young's modulus value is about 83.17 GPa, which is highly compatible with the literature experimental data (i.e., $89.35 \pm 2.2$ GPa under the strain rate of $3 \times 10^{-4}$ 1/s) measured by Song et al. [13] using nanoindentation. This Young's modulus value is relatively low in comparison with other IMCs widely used in microelectronics packaging, such as $Cu_6Sn_5$, $Cu_3Sn$, $Cu_5Zn_8$, $Ni_3Sn_4$, $AuZn_3$, and $Cu_{11}In_9$ [13] (see Table 3).

To identify the brittleness or ductility of a material, Pugh [47] proposed a criterion based on the $K/G$ ratio. The threshold value for the $K/G$ ratio is 1.75: if the $K/G$ ratio exceeds the threshold value, the material possesses the ductile nature; otherwise, a brittle one. The calculated $K/G$ ratio of the $Ag_9In_4$ crystal is also presented in Table 3, and the $K/G$ ratio is superior to 1.75, revealing that $Ag_9In_4$ is a ductile material. This result is a good match with the prediction based on the Pettifor criterion of Cauchy pressure, as shown in Equation (14). Thus, it is well believed that due to its ductile nature, this IMC possesses good impact resistance and thus can potentially enhance the drop impact solder interconnect reliability of microelectronic packages [48]. The hardness of a material usually plays a crucial role in abrasive-wear-resistant applications. Teter [49] proposed a linear relationship between the Vickers hardness $H_v$ and the shear modulus $G$ if a material is intrinsically brittle. Noteworthy is that material hardness is also linked to both shear modulus $G$ and bulk modulus $K$ for various materials. Chen et al. [50] introduced an improved non-linear correlation in terms of both $K$ and $G$. It is, however, found that this correlation may lead to a negative Vickers hardness as a result of the last correlation term

"$-3$". Later on, Tian et al. [51] introduced a modified correlation model that can ease this concern,

$$H_v = 0.92\alpha^{1.137}G^{0.708}, \alpha = G/K \tag{23}$$

It was reported that materials with a Vickers hardness greater than 40 GPa can be categorized as a superhard material [52]. The calculated value of $H_v$ for $Ag_9In_4$ is 3.67 GPa, as shown in Table 3, which is far less than the threshold value, implying that $Ag_9In_4$ may be considered as a low densification material. The calculated Vickers hardness of the $Ag_9In_4$ crystal is also in line with the literature experimental data [13] (i.e., $3.18 \pm 0.22$ GPa under the strain rate of $3 \times 10^{-4}$ 1/s). Table 3 further demonstrates the comparison of Vickers hardness between $Ag_9In_4$ and other commonly found IMCs in microelectronics packaging, such as $Cu_6Sn_5$, $Cu_3Sn$, $Cu_5Zn_8$ and $Cu_{11}In_9$ [13]. It is clear to see that the $Ag_9In_4$ is a material with a relatively low hardness, and the relatively low Vickers hardness may be pertinent to the weak metal bond.

### 3.3. Characterization of Elastic Anisotropic Properties

One of the most important issues for a material is the elastic anisotropy, primarily determining the bonding nature in various crystallographic directions. Essentially, this elastic parameter would greatly affect the materials' physical properties, such as elastic instability, anisotropic plastic deformation, and crack behavior. Previously, the study of elastic anisotropy of a material has been well-developed in the physics of crystal. The elastic anisotropy can be further depicted using the universal anisotropic index $A^U$ [53] and the percentage elastic anisotropy in compressibility and shear, i.e., $A^B$ and $A^G$ [54,55], respectively,

$$A^U = 5\frac{G_V}{G_R} + \frac{K_V}{K_R} - 6 \tag{24}$$

$$A^B = \frac{K_V - K_R}{K_V + K_R} \times 100\% \tag{25}$$

$$A^G = \frac{G_V - G_R}{G_V + G_R} \times 100\% \tag{26}$$

These values range from zero, representing an isotropic material, to 1 (100%), denoting the maximum anisotropy. The calculated results are displayed in Table 4, where the $A^U$ value of $Ag_9In_4$ is 0.088, implying elastic anisotropy. Furthermore, the zero value of $A^B$ indicates elastic isotropy in its compression behavior, while the value of 0.87 for $A^G$ reveals shear anisotropy (%).

**Table 4.** The calculated anisotropic indices of the $Ag_9In_4$ crystal.

| Anisotropic Index | $A^U$ | $A^B$ (%) | $A^G$ (%) |
|:---:|:---:|:---:|:---:|
| Value | 0.088 | 0 | 0.87 |

The 3D surface representation of the Young's modulus of $Ag_9In_4$ and its two-dimensional (2D) projections onto the $yz$, $xz$, and $xy$ planes are given in Figure 3, respectively. The degree of anisotropy depends on the deviation of a geometrical body from the spherical shape. If a geometrical body is a sphere, it presents isotropy. Therefore, the directional Young's modulus showed the $Ag_9In_4$ holds the anisotropic characteristic. In addition, the Young's modulus of $Ag_9In_4$ had a maximum value of 95.78 GPa in the <001> crystal direction and a minimum value of 75.56 GPa in the <111> crystal direction. The maximum and minimum Young's moduli are listed in Table 5.

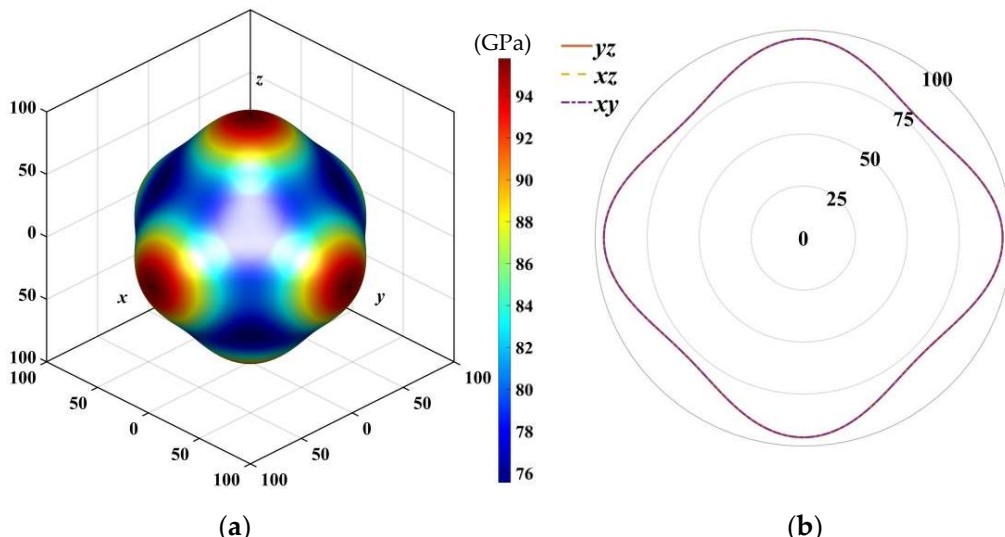

**(a)**            **(b)**

**Figure 3.** (**a**) 3D surface representation of Young's modulus of Ag$_9$In$_4$ and (**b**) 2D projections of Young's modulus onto *yz, xz,* and *xy* planes.

**Table 5.** Maximum and minimum Young's moduli for Ag$_9$In$_4$ crystal.

| Young's Modulus | Whole | *yz* | *xz* | *xy* |
|---|---|---|---|---|
| $E_{\max}$ (GPa) | 95.78 | 95.78 | 95.78 | 95.78 |
| $E_{\min}$ (GPa) | 75.56 | 79.78 | 79.78 | 79.78 |

Generally, the maximum and minimum values in each direction are used to evaluate the directional shear modulus. Figures 4 and 5 display the 3D surface and its 2D projections onto the *yz, xz,* and *xy* planes of Ag$_9$In$_4$. Similar to the Young's modulus, there is an anisotropic nature in the shear modulus in the three crystal planes. Furthermore, the maximum shear modulus of Ag$_9$In$_4$ was maximum in the <110> crystal direction and minimum in the <001> crystal direction. On the other hand, the minimum shear modulus was maximum in the <111> crystal direction and minimum in the <001> crystal direction.

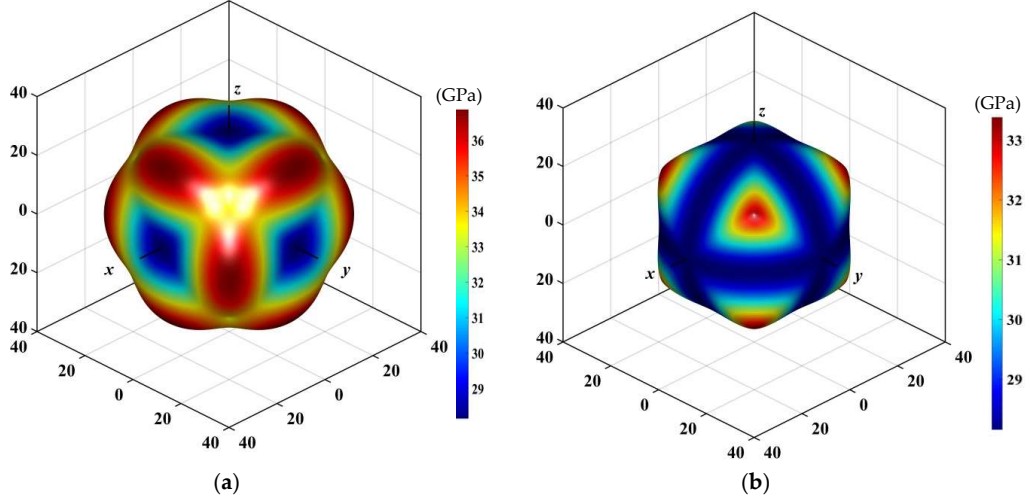

**(a)**            **(b)**

**Figure 4.** Three-dimensional surface representation of (**a**) maximum shear modulus and (**b**) minimum shear modulus of Ag$_9$In$_4$.

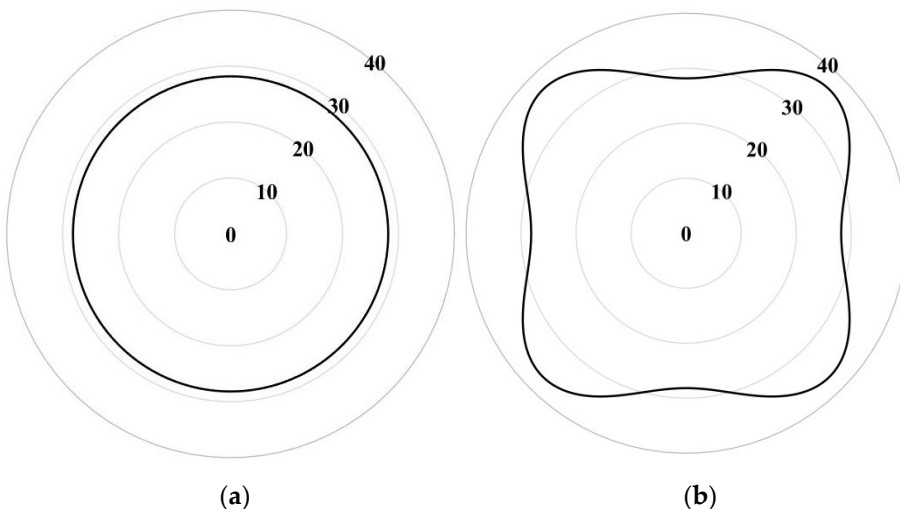

**Figure 5.** Two-dimensional projections in *yz*, *xz* and *xy* planes of (**a**) maximum shear modulus and (**b**) minimum shear modulus of Ag$_9$In$_4$.

The maximum and minimum values of the shear moduli are listed in Table 6.

**Table 6.** Maximum and minimum values for the shear moduli of Ag$_9$In$_4$.

| Shear Modulus | Whole | *yz* | *xz* | *xy* |
|---|---|---|---|---|
| $G_{max}$ (GPa) | 36.86 | 28.16 | 28.16 | 28.16 |
| $G_{min}$ (GPa) | 28.16 | 28.16 | 28.16 | 28.16 |

### 3.4. Thermodynamic Properties

The predicted Debye temperature for Ag$_9$In$_4$ is approximately 201.4K at zero pressure and 300K. The temperature-dependent heat capacity is exhibited in Figure 6. It indicates that the heat capacity grows markedly with the increase in temperature from 0K to the Debye temperature. In addition, by performing the curve fitting process at temperatures less than the Debye temperature, the heat capacity presents a power-law temperature relationship with a power exponent of around 3 (i.e., 2.96). This power-law exponent fully coincides with the so-called Debye heat capacity theory, which infers that at constant volume, the low temperature heat capacity is proportional to T$^3$ [56]. At 300K, the predicted heat capacity is 25.3 J/(mol·K), and at temperatures greater than the Debye temperature, it is converging to around 25.9 J/(mol·K), which is the classical Dulong–Petit value [57]. In general, the Dulong–Petit value appears in solids at temperatures way above the Debye temperature. Additionally, it was conspicuous that the heat capacity became constant at high temperature. This indicated that the phonons do not interact with each other, thus leading to an independent relationship between crystal volume and temperature. Therefore, the present approach could not evaluate the thermal expansion coefficient, which describes the relationship between volume change and temperature variation.

The thermal conductivity of a material tends to decrease with temperature and ultimately reaches a limiting magnitude, which is termed the minimum thermal conductivity or high-temperature thermal conductivity. For a high-temperature application, a good estimate of the material's minimum thermal conductivity is crucial. The minimum thermal conductivity can be described using Cahill et al.'s equation [58],

$$k_{mim} = \frac{k_B}{2.48}(p)^{\frac{2}{3}}(v_l + 2v_t) \tag{27}$$

where $k_B$ stands for the Boltzmann constant, $p$ denotes the number of atoms per unit volume, and $v_l$ and $v_t$ represent the longitudinal and transverse sound velocities, respectively. The $v_l$ and $v_t$ can be expressed as follows,

$$v_l = [(K + 4G/3)/\rho]^{\frac{1}{2}} \tag{28}$$

$$v_t = [G/\rho]^{\frac{1}{2}} \tag{29}$$

where $\rho$ represents the material density. Table 7 illustrates the computed longitudinal sound velocity, transverse sound velocity, and minimum thermal conductivity. Substituting the calculated longitudinal and transverse sound velocities into Equation (27) gives the minimum thermal conductivity of 0.544 W/(m·K) at zero pressure.

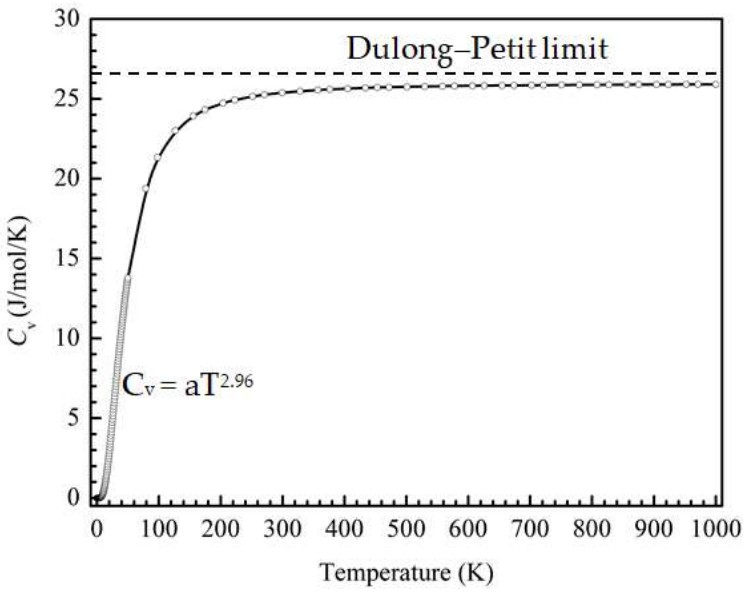

**Figure 6.** Heat capacity of $Ag_9In_4$ as a function of temperature.

**Table 7.** Mass density, longitudinal, transverse sound velocities and minimum thermal conductivity of $Ag_9In_4$.

| Property | $\rho$ (Kg/m³) | $v_l$ (m/s) | $v_t$ (m/s) | $k_{min}$ (W/(m·K)) |
|:--------:|:--------------:|:-----------:|:-----------:|:-------------------:|
| Value | 9389.8 | 3594.2 | 1827.8 | 0.544 |

## 4. Conclusions

This study applied ab initio DFT simulations to characterize the physical properties of the cubic $Ag_9In_4$ crystal. Their relationship with crystalline direction and temperature was investigated. The calculated lattice constants, Young's modulus, and Vickers hardness turned out to be in great conformity with the experimental results from the literature, indicating the feasibility of the present theoretical predictions. The theoretical calculations demonstrated that $Ag_9In_4$ possesses structural and mechanical stability and ductility, thereby possessing good impact resistance. In addition, $Ag_9In_4$ is a material with a relatively low stiffness and hardness, as compared to other broadly found IMCs, including $Cu_6Sn_5$, $Cu_3Sn$, $Cu_5Zn_8$, and $Cu_{11}In_9$, and elastic anisotropy. Specifically, $Ag_9In_4$ reveals an anisotropic characteristic in the shear modulus and Young's modulus but an isotropic nature in compressibility and the bulk modulus. At last, the heat capacity of $Ag_9In_4$ at temperatures below the Debye temperature obeys the $T^3$-law, and that at temperatures above the Debye temperature follow the classical Dulong–Petit rule.

**Author Contributions:** Conceptualization, H.-C.C. and C.-F.Y.; data curation, C.-F.Y.; formal analysis, C.-F.Y.; investigation, C.-F.Y.; software, C.-F.Y.; validation, H.-C.C. and C.-F.Y.; visualization, C.-F.Y.; writing—original draft, C.-F.Y.; writing—review and editing, H.-C.C. and C.-F.Y.; funding acquisition, H.-C.C.; project administration, H.-C.C. All authors have read and agreed to the published version of the manuscript.

**Funding:** This research was funded by National Science and Technology Council, Taiwan, under grants MOST 109-2221-E-035-004-MY3 and MOST 110-2221-E-035-049-MY3.

**Institutional Review Board Statement:** Not applicable.

**Informed Consent Statement:** Not applicable.

**Data Availability Statement:** Data sharing not applicable.

**Conflicts of Interest:** The authors declare no conflict of interest.

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
