# Peer review of "A DFT Characterization of Structural, Mechanical, and Thermodynamic Properties of Ag9In4 Binary Intermetallic Compound"

_metals, doi:10.3390/met12111852_

Round 1

Reviewer 1 Report

Dear authors,

You have done a good job.

Reviewer 2 Report

The present manuscript is a representative of a frequent type of works, in which DFT-based electronic total energy calculations are performed on some selected crystalline material and where ground state structure as well as other properties are derived from the resulting energies (e.g. elastic constants but also T > 0 structure and thermodynamic properties). Such papers can by of largely varying properties. Such papers can be of good value if real questions or trends (e.g. between different materials) are adressed. Many of these paper produce little real insight but, in contrast mainly fill the space by applying routine calculations, e.g. based on the single-crystal elastic constants.
The present paper singles out the Ag9In4 gamma-brass phase relevant for some soldering applications. This material has the well-established I-type gamma brass structure. The authors determine the static ground state structure by relaxed calculations, they calculate further the single-crystal elastic constants and they appear to estimate the phonon-based T > 0 K contribution to the Gibbs energy, by not considering the phonon dispersion but considering only the gamma-point properties contained in the elastic constants (as much as I understand it on the fly).
Unfortunately, the current manuscript is on the very poor side of the range of qualities I have described in my first paragraph. Two characteristics can be identified for the current manuscript:
1.    There is severe and in my view scientifically justified self-citation of papers by the first author. I only detail the early self-citations in the introduction. Instead of introducing the topic by general papers (e.g. reviews) related to the topic, the authors find excuses to cite a series of very specialized papers of their own.
2.    While the DFT-calculations on Ag9In4 may be regarded as routine and are likely performed with adequate methodology leading to correct results. The calculations performed based on e.g. the single-crystal elastic constants e.g. representation of the elastic moduli and the Poisson ratio, however, strongly imply that the authors do not understand the physical meaning of the formulas which they use.
An incomplete list of details:
(i)    Figure 1 and 2 do not add anything to the paper.
(ii)    Line 51 under bump METALLIZATION (here it is the adequate term)
(iii)    Line 90 [26-34]: I checked a couple of these papers. Again this inadequate self citation, without adding content to the manuscript! Moreover, at least some of these papers are not nanomaterial. A material like Pd2Al might be prepared in nanoscopic form, but it is not a nanomaterial by itself, and as I can see it from checking the paper: Nanoscopic properties are not addressed in the cited paper.
(iv)    Line 112: [C_ij] is technically not a tensor, because when using the Voigt notation (which I assume that it is the case) the 6 x 6 matrix does not reflect transformation properties.
(v)    Formula 3,4 and below: use either lower- or upper-case S to indicate the indices. s_0 in Formula 5 has not been defined.
(vi)    Line 123: theta and phi are not Euler angles!
(vii)    Formulas 5,6 might have been used in the literature (I cannot check all this) but they cannot be valid as described: For a shear modulus you need three independent defining directions: two define a plane, and a third one to define the shear direction within that plane.
(viii)    Formula 7: Now the angle theta has a different meaning than in Formula 6, but this has not been considered. Now, however, the three-parameter character of the Poisson ratio (as for G) has been reflected in the formula. It is, however, not clear to how the three parameter characteristics is reflected in Figure 7 (I also believe that the authors also do not understand what they show).
(ix)    General: What is the source of the atomic coordinates for the Ag9In4 structure. Apparently no coordinates were given in Ref. [49]
(x)    Formula 8: I assume that E(V) is NOT the Young’s modulus from Eq. (4)
(xi)    Formula 13: v is likely Hill-average of the Poisson ratio not yet defined (this happens later)
(xii)    Figure 3: This is not a self-contained figure caption
(xiii)    The considerations based on Eq. (27-33)  are totally irrelevant, because the outcome of these formulas directly simplifies using the cubic symmetry (see Eq. (2)). This indicates that A1 = A2 = A3, A_B = 0, Aba = Abc =1 for all cubic crystals. This is a waste of space, and even if you pay for this, such space should not be wasted in a scientific journal.
(xiv)    For the same reason, delete Figure 4. This will be a sphere for all cubic crystals.
(xv)    Delete Figure 8
(xvi)    Figure 9 does not show a visible T^3 range as claimed in the text.
Here I stop. The authors really should reconsider their way of scientific working.

Round 2

Round 3

Reviewer 2 Report

1. After the authors appear to cite an appropriate software I have to go back to figure 4, where the authors have attempted to give some representation of the shear modulus resulting from the elastic constants. Now the authors seem to have acknowledged that the shear modulus is a function of three parameters, shear plane (two) and direction within the plane (one parameter). But still figure 4 suggests that that it would be a function which depends like the Young’s modulus on a single direction. This must be clarified: What is exactly shown and what is the relation of the shown graphics with the anisotropic shear modulus. This must be made clear, and I am not willing to give hints.

2. At the same time I have to realise that also Figure 3 is not appropriately described. What is exactly a 3D surface representation of the Young’s modulus and what is the meaning of all the colors and numbers?

3. The anisotropy indices A_B and A_G have the unit in % in Table 4

4. As I get fed up, line 114 etc have to read

... by generalized Hooke’s law in Voigt notation

....

where {sigma_i} and {eps_j} denote the stress vectors, respectively....

5. upper case “S” in Eqs. 16, 18.
